# The Conundrum of Systemic Arterial Pressure Management on Cardiopulmonary Bypass

**DOI:** 10.3390/jcm12030806

**Published:** 2023-01-19

**Authors:** Marco Ranucci, Mauro Cotza, Umberto Di Dedda

**Affiliations:** Department of Cardiothoracic and Vascular Anesthesia and Intensive Care, IRCCS Policlinico San Donato, 20097 Milan, Italy

## 1. Introduction

The recently released EACTS/EACTA/EBCP guidelines on cardiopulmonary bypass (CPB) in adult cardiac surgery [1] address, within the general topic, the management of systemic arterial pressure (SAP) during CPB. The recommendation (Class I A) is to maintain the SAP between 50 and 80 mmHg by using vasodilators or vasoconstrictors to remain within this range. This reflects the general standard of clinical management, and could be interpreted as a general rule. However, this range is quite large, and does not consider specific needs of different patients or categories of patients. Within the concept of “precision medicine”, and as quoted in the same guidelines, attempts to individualize the optimal SAP during CPB appear reasonable. The present review addresses the basic physiology of SAP during CPB and the pathophysiological factors impacting SAP; the existing literature with respect to patient outcomes at different levels of SAP; and the possible strategies to individualize SAP management during CPB.

## 2. Basic Physiology of SAP during CPB

The majority of CPB surgeries are presently managed with nonpulsatile flow. This means that during the aortic cross-clamp, the SAP is maintained at a constant level and is, therefore, equivalent to the mean systemic arterial pressure (MAP). There are exceptions to this pattern of flow, represented by the pulsatile flow obtained with specific measures applied to the pump, or by the use of CPB without aortic cross-clamping. Although both of these techniques are certainly of interest, they are applied in a minority of cases and institutions, and will not be addressed in the present review.

MAP during CPB follows the general rule of the Poiseuille law, a physical law that determines the pressure drop in an incompressible and Newtonian fluid in laminar flow flowing through a long cylindrical pipe of a constant cross-section. Although blood is a non-Newtonian fluid, this law can be applied, with some adaptations, to the human circulation and to CPB conditions.

Basically, the Poiseuille law states that:
$$\Delta P=\frac{Q\times 8\eta L}{{\pi R}^{4}}$$
where ΔP is the pressure gradient required to generate the flow (Q), η is the dynamic viscosity of the fluid, L is the length of the pipe, and R is the radius of the pipe. The length of the pipe, the radius of the pipe, and the fluid viscosity are components of the resistance to flow. Therefore, the Poiseuille law is a derivate of the general concept that, at a constant flow, increasing the resistance increases the pressure.

Within the concept of human circulation in natural conditions and during CPB, the resistive component can be separated into a circuit component (the systemic vascular resistances to blood flow that are affected by systemic vasodilation or vasoconstriction) and a fluidic component (the blood viscosity that is affected by the hematocrit and the temperature).

### 2.1. The Fluidic Component: Blood Viscosity

Blood viscosity depends on the hematocrit (HCT) and the blood temperature. The impact of the HCT on the dynamic blood viscosity is typical of a non-Newtonian fluid. Given its corpuscular component (of which the red blood cells are the main components), the flow is of a nonlaminar nature, and the higher the HCT levels, the higher the dynamic blood viscosity levels. However, this relationship is nonlinear (Figure 1). Conversely, with respect to the impact of temperature, the blood reacts like a Newtonian fluid, with the dynamic viscosity linearly increasing as the temperature decreases (Figure 2).

Both these factors are of peculiar relevance during CPB circulation because hemodilution and hypothermia are common features in clinical practice. At a normal HCT of 40%, the dynamic blood viscosity is about 4 centipoise (cP), and it decreases by about 25% at an HCT of 30% and by about 45% at an HCT of 20%. These effects are only partially counteracted by hypothermia because the dynamic viscosity increases by about 2% for each 1-centigrade decrease. Therefore, even if hemodilution is limited to an HCT of 30%, the dynamic blood viscosity will be decreased by 20% under moderate hypothermia (30–32 °C), and will be totally restored at a blood temperature around 22 °C (profound hypothermia).

Therefore, in general, it can be assumed that during CPB the fluidic component of resistance decreases, creating the environment for a decrease in MAP.

### 2.2. The Circuit Component: The Systemic Peripheral Vascular Resistances

Under CPB conditions and nonpulsatile flow, there are factors increasing and factors decreasing the peripheral systemic vascular resistances. Nonpulsatile flow has a direct effect in increasing the peripheral vascular resistances due to an increased level of angiotensin II. Peripheral vascular resistance index levels that were increased by more than 50% have been reported, and this increase was blunted when pulsatile perfusion was applied [2]. 

On the other side, CPB and contact with the foreign surfaces of the circuit and oxygenator induce a release of a number of cytokines that result in profound vasodilation [3]. The preoperative use of a number of medications, namely, angiotensin-converting enzyme inhibitors, are associated with vasoplegic syndrome during CPB [4]. Depending on the balance between vasoconstricting and vasodilating stimuli, patients under CPB are prone to both hypotension and hypertension.

## 3. The Effects of MAP on CPB As Determinants of Outcomes in Cardiac Surgery

### 3.1. Retrospective Studies

A considerable number of studies addressed the impact of MAP on CPB with respect to clinical outcomes. According to Reich and associates [5], patients with an MAP < 50 mmHg had a 30% increase in in-hospital mortality, and Sun and associates found that postoperative stroke was significantly more frequent in patients experiencing an MAP < 64 mmHg, with a 13% increase in risk for every 10 min between 55 and 64 mmHg, and a 16% increase for every 10 min < 55 mmHg [6]. These and other studies suffer from the usual problems of large retrospective studies. They tell us that patients that spontaneously experience a relatively low MAP during CPB may suffer from bad outcomes after surgery. However, they cannot adjust for a number of confounders, i.e., the reaction of clinicians to a low MAP (use of vasoconstrictors), and the mechanisms leading to a low MAP. Among these, it must be underlined that pump flow and HCT are important determinants of MAP during CPB, as they concur in determining oxygen delivery. Low levels of oxygen delivery are associated with bad outcomes [7,8] and low levels of HCT per se are associated with a number of adverse events [9,10]. Since low levels of HCT are determinants of low viscosity and consequent hypotension, it is possible that the association between low MAP and bad outcomes found in retrospective studies may simply reflect the effects of hemodilution and low oxygen delivery.

### 3.2. Prospective Studies

Presently, there have been eight randomized controlled trials (RCTs) addressing the effects of low vs. high MAP during cardiac surgery with CPB [11,12,13,14,15,16,17,18]. Some of these studies investigated the effects of different MAPs before, during, and after CPB [16], but the majority addressed only different MAP regimens during CPB. However, the definitions of “low” or “high” MAP during CPB greatly varied across the different studies. A “low” MAP during CPB was defined as 40–50 mmHg [17], 50–60 mmHg [12,14,16,18], and <70 mmHg [11,13,15]. A “high” MAP was defined as >60 mmHg [18], >70 mmHg [11,14], 70–80 mmHg [17], 75–85 mmHg [16], 70–90 mmHg [15], 80–90 mmHg [13], and 80–100 mmHg [12]. This wide heterogeneity, especially in the definition of the “high” MAP, introduces a bias into the interpretation of the results. Additionally, different interventions were applied to reach and maintain the target MAP, even if, in general, they were based on the use of vasodilators and vasoconstrictors.

The outcome measures again differ among the different RCTs. Basically, the most studied negative outcomes were stroke/neurocognitive decline [12,13,14,15,17,18], acute kidney injury [11,13,14,16,18], and intensive care unit and hospital stay [12,13,14,15,16,17]. The results of these RCTs are quite heterogeneous. There is a general agreement that perioperative stroke incidence and neurocognitive outcomes do not change with different regimens of MAP, with only one study showing a larger decline in postoperative neurocognitive function in the low MAP group [13]. Myocardial infarction and acute kidney injury showed no significant differences between the low and high MAP groups in all the studies that addressed these outcomes. The same applies to hospital/operative mortality.

The eight RCTs were pooled together in a meta-analysis by McEwen and associates [19]. The authors had a difficult time navigating through the high heterogeneity of the studies and the low quality of some of them. However, they concluded that no outcome measures demonstrated any difference between low and high MAP groups, with the only exception being a significantly higher risk of transfusion in the high-target group (relative risk 1.4, 95% confidence interval 1.1–1.9, *p* < 0.01).

### 3.3. The New Research Areas: Toward a Personalized Approach

The concept of the “optimal” perfusion pressure during CPB is present in the environment of personalized medicine. It is quite obvious that different patients with different ages, different vascular conditions, and different organ functions may have the same “best MAP”. When searching for an individual’s best MAP (or best range of MAP) during CPB, the main physiological concept to respect is that changes in MAP should not determine changes in peripheral organ perfusion, namely, to the brain, kidney, and visceral organs. This means that the MAP should stay within the limits of flow “autoregulation”.

The most widely studied organ, in this respect, is the brain. Due to its peculiar nature and high susceptibility to MAP changes, the brain offers an almost perfect model for autoregulation. Additionally, cerebral blood flow (CBF, measured in the middle cerebral artery) can be easily measured with Doppler ultrasound technology. Another tool linking the MAP to the brain blood flow is near-infrared spectroscopy (NIRS). Actually, NIRS is not a direct flow measure, but rather is a marker of brain oxygen extraction. However, under steady conditions, cerebral regional oxygen saturation determined via NIRS may be considered as a flow-dependent parameter.

There are studies introducing the concept of a “personalized” MAP range during CPB that is based on the limits of autoregulation. Basically, the technique to detect the lower limit of autoregulation is based on coupling the MAP and an indirect (NIRS) [20,21,22] and/or direct (brain Doppler) [20,22,23,24,25] measure of CBF. Spontaneous variations of MAP are utilized, and the autoregulation range corresponds to the MAP values where no correlation (moving the Pearson correlation coefficient) between the MAP and the correspondent measure of CBF exists.

Some of these studies simply addressed the correlation between the time spent during CPB below or above the lower and upper MAP limits of autoregulation. Liu and associates [20] found that the area under the curve of an MAP less than the lower limit of autoregulation was independently associated with postoperative acute kidney injury, major morbidity, and mortality. Hori and associates [21] found that patients with postoperative delirium had a higher rate of MAP excursions above the optimal MAP. Conversely, the same group observed that the duration and magnitude of an MAP below the lower limit of autoregulation were associated with the risk of stroke [23].

There are RCTs that compared the standard MAP target strategy to a strategy based on a target within the limits of autoregulation. Hogue and associates [24] could not find any difference in the risk of stroke or new lesions with magnetic resonance; conversely, delirium occurred in 8.2% of patients treated according to the cerebral autoregulation-based MAP target vs. 14.9% of patients in the standard MAP target group (risk ratio 0.55, 95% confidence interval 0.32–0.93, *p* = 0.035), and the treated patients showed an improved performance in the memory test 4–6 weeks after surgery. Brown and associates [25] confirmed that delirium was less frequent in patients treated according to the cerebral autoregulation-based MAP target (risk ratio of 0.55, 95% confidence interval 0.31–0.97, *p* = 0.04).

## 4. Future Research Areas and Conclusions

All the existing studies have focused on clinically relevant primary endpoints, such as renal dysfunction, brain injury, and mortality. It would be interesting to include, in future studies, more sensitive biomarkers of organ dysfunction. Among many possible biomarkers, arterial blood lactates are a simple and easily available measure of the adequacy of peripheral perfusion with respect to oxygen needs. Hyperlactatemia during CPB has been associated with a number of negative outcomes [26], and although an inadequate MAP could be only one of the possible reasons for hyperlactatemia, the effects of different MAP regimens on arterial blood lactates deserve attention.

The present state-of-the-art research cannot claim the superiority of any level of MAP within the range of 50 mmHg to 80 mmHg, suggesting that attempts to force the MAP toward higher levels during CPB by extensively using vasoconstrictors, rather than being beneficial, could lead to adverse outcomes. Conversely, the use of vasoconstrictors to correct abnormally low MAP values (vasoplegic syndrome) is recommended [1]. A very recent meta-analysis on three randomized controlled trials confirms that no clinically relevant changes were found for high vs. low MAP targets during CPB [27]. The perspective of an individualized “best MAP range” assessment based on cerebral autoregulation is certainly interesting; however, it must be considered that this approach is based on complex computerized algorithms, specific technology, and local expertise. Future studies should try to detect the autoregulation range in an awake patient, coupling MAP and CBF by inducing blood pressure variations and not spontaneous intraoperative MAP changes.

## Figures and Tables

**Figure 1 jcm-12-00806-f001:**
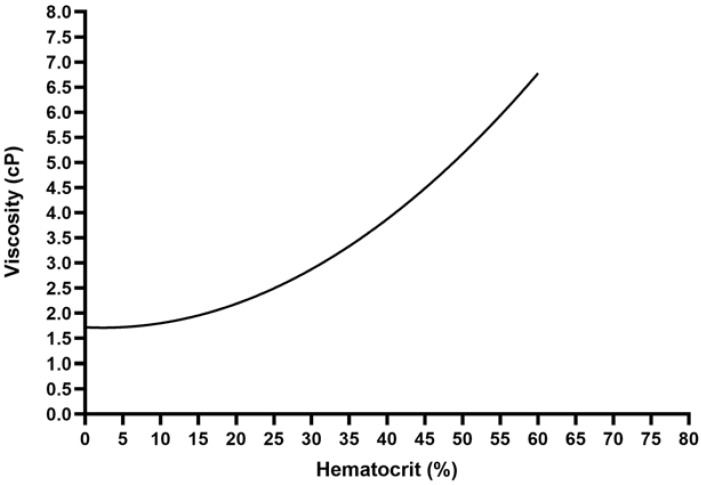
Relationship between hematocrit and blood viscosity.

**Figure 2 jcm-12-00806-f002:**
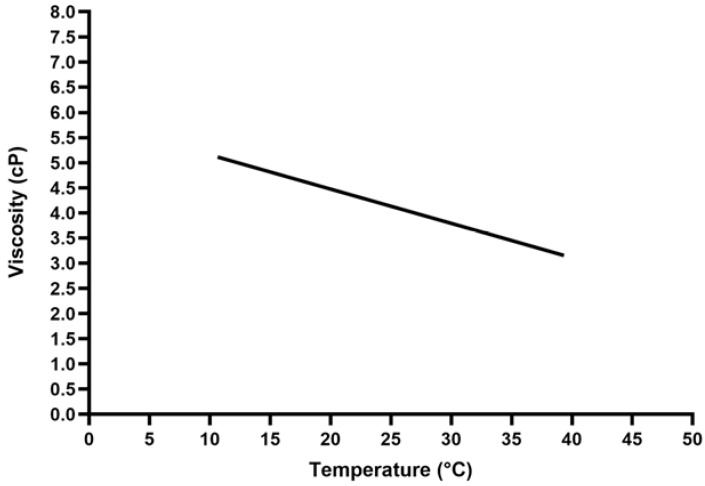
Relationship between temperature and viscosity.

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
