# Peer review of "The Conundrum of Systemic Arterial Pressure Management on Cardiopulmonary Bypass"

_jcm, 2023, doi:10.3390/jcm12030806_

Round 1

Reviewer 1 Report

Marco Ranucci et al performed a narrative review of the critics surrounding the appropriate threshold of systemic and mean arterial pressure (SBP and MAP) in cardiopulmonary bypass (CPB) and its physiological consequences. The authors give us a review of the basic physiology of SAP on CPB, followed by their clinical relevance on clinical outcomes and final perspectives surrounding some controversies and future areas of opportunity for research. The present review is of interest for physicians in critical care and internal medicine. I have some minor comments.

Figures should include a figure legend. Moreover, the diagrams used were based on previously published evidence? Clarify

Please avoid using subjective words at the interpretation, such as “nice”, “well-performed study”, “fascinating”,etc.

What does the authors refer to “moving Pearson correlation coefficient

Author Response

See the attached word file

Reviewer 2 Report

It is difficult to define the indicated blood pressure during CPB management. This paper provides a detailed description of circulatory management on CPB. During intraoperative management, it is important to avoid complications of organ ischemia and organ damage. Lactate levels are an acute indicator of organ injury. In this review article, a note on lactate is missing. The author should add a note on lactate levels during CPB.

Reviewer 3 Report

This paper nicely reviewed the literature about the management of systematic arterial pressure (SAP) during cardiopulmonary bypass (CPB). I have some minor comments.

1. Yuki Kotani et al. also did a meta-analysis to evaluate the benefits and harms of higher versus lower blood pressure targets during cardiac surgery with CPB (PMID: 36448514). They only included three RCTs in their paper. The authors may consider discussing this paper in the discussion.

2. Is there any ongoing clinical studies the authors found in their search that aim to investigate the relationship between SAP during CPB and outcomes?

3. Please add figure legends.

Round 2

Reviewer 2 Report

The paper has been appropriately revised according to my remarks.